# Guidelines to Analyze Preclinical Studies Using Perinatal Derivatives

**DOI:** 10.3390/mps6030045

**Published:** 2023-04-25

**Authors:** Ana Salomé Pires, Sveva Bollini, Maria Filomena Botelho, Ingrid Lang-Olip, Peter Ponsaerts, Carolina Balbi, Anna Lange-Consiglio, Mathilde Fénelon, Slavko Mojsilović, Ekaterine Berishvili, Fausto Cremonesi, Maria Gazouli, Diana Bugarski, Alexandra Gellhaus, Halima Kerdjoudj, Andreina Schoeberlein

**Affiliations:** 1Coimbra Institute for Clinical and Biomedical Research (iCBR) Area of Environment, Genetics and Oncobiology (CIMAGO), Institute of Biophysics, Faculty of Medicine, University of Coimbra, 3000-548 Coimbra, Portugal; 2Center for Innovative Biomedicine and Biotechnology (CIBB), University of Coimbra, 3000-548 Coimbra, Portugal; 3Clinical Academic Center of Coimbra (CACC), 3000-354 Coimbra, Portugal; 4Department of Experimental Medicine (DIMES), University of Genova, 16132 Genova, Italy; 5Division of Cell Biology, Histology, Embryology, Gottfried Schatz Research Center, Medical University of Graz, 8010 Graz, Austria; 6Laboratory of Experimental Hematology, Vaccine and Infectious Disease Institute (Vaxinfectio), Faculty of Medicine and Health Sciences, University of Antwerp, 2610 Antwerp, Belgium; 7Laboratory of Cellular and Molecular Cardiology, Istituto Cardiocentro Ticino, CH-6900 Lugano, Switzerland; 8Center for Molecular Cardiology, University of Zurich, CH-8057 Zurich, Switzerland; 9Department of Veterinary Medicine and Animal Science (DIVAS), Università degli Studi di Milano, Via Celoria, 10, 20133 Milano, Italy; 10INSERM U1026, University of Bordeaux, Tissue Bioengineering (BioTis), F-33076 Bordeaux, France; 11CHU Bordeaux, Service de Chirurgie Orale, F-33076 Bordeaux, France; 12Group for Hematology and Stem Cells, Institute for Medical Research, University of Belgrade, 11000 Belgrade, Serbia; 13Laboratory of Tissue Engineering and Organ Regeneration, University of Geneva, CH-1211 Geneva, Switzerland; 14Department of Basic Medical Sciences, Laboratory of Biology, Faculty of Medicine, School of Health Science, National and Kapodistrian University of Athens, 115 27 Athens, Greece; 15Department of Gynecology and Obstetrics, University Hospital Essen, University of Duisburg-Essen, D-45147 Essen, Germany; 16Biomatériaux et Inflammation en Site Osseux (BIOS), Université de Reims Champagne Ardenne, F-51097 Reims, France; 17Department of Obstetrics and Feto-maternal Medicine, Inselspital, Bern University Hospital, University of Bern, CH-3010 Bern, Switzerland; 18Department for BioMedical Research (DBMR), University of Bern, CH-3008 Bern, Switzerland

**Keywords:** perinatal derivatives, preclinical studies, animal models, database search, consensus, protocol

## Abstract

The last 18 years have brought an increasing interest in the therapeutic use of perinatal derivatives (PnD). Preclinical studies used to assess the potential of PnD therapy include a broad range of study designs. The COST SPRINT Action (CA17116) aims to provide systematic and comprehensive reviews of preclinical studies for the understanding of the therapeutic potential and mechanisms of PnD in diseases and injuries that benefit from PnD therapy. Here we describe the publication search and data mining, extraction, and synthesis strategies employed to collect and prepare the published data selected for meta-analyses and reviews of the efficacy of PnD therapies for different diseases and injuries. A coordinated effort was made to prepare the data suitable to make statements for the treatment efficacy of the different types of PnD, routes, time points, and frequencies of administration, and the dosage based on clinically relevant effects resulting in clear increase, recovery or amelioration of the specific tissue or organ function. According to recently proposed guidelines, the harmonization of the nomenclature of PnD types will allow for the assessment of the most efficient treatments in various disease models. Experts within the COST SPRINT Action (CA17116), together with external collaborators, are doing the meta-analyses and reviews using the data prepared with the strategies presented here in the relevant disease or research fields. Our final aim is to provide standards to assess the safety and clinical benefit of PnD and to minimize redundancy in the use of animal models following the 3R principles for animal experimentation.

## 1. Introduction

Perinatal derivatives (PnD) have been widely recognized as therapeutic tools for tissue engineering and regenerative medicine applications in a broad field of diseases or injuries in the past few years. The term PnD includes birth-associated tissues obtained from term placentas and fetal annexes in their naïve or processed form such as the amniotic and chorionic membranes, chorionic villi, the umbilical cord, the basal plate, and the amniotic fluid, as well as the cells isolated from these tissues and the factors released by them (such as free nucleic acids, soluble proteins, lipids, extracellular vesicles, or extracellular matrix components) [1]. Preclinical studies are an essential milestone on the road to clinical translation. There has been extensive research using PnD therapy in animal models, and publications describing in vivo preclinical studies using PnD have shown annual increases in the last 10–15 years. The perinatal derivatives studied are diverse regarding donor population, tissue source, isolation, expansion, and storage protocols. The study designs are equally heterogeneous: even for the same target disease, outcome measures include various parameters and are often evaluated at different time points [2]. Therefore, it is unclear which types of PnD provide optimal therapeutic results, when and how they are best applied, and which are the underlying mechanistic effects of therapy using PnD.

The International Network for Translating Research on Perinatal Derivatives into Therapeutic Approaches (SPRINT, CA17116) (https://www.sprint-cost.org/) (accessed on April 2023), funded by COST (European Cooperation in Science and Technology) (https://www.cost.eu/), (accessed on April 2023) is an initiative to bridge the gaps between PnD research and their translation into the clinic. Specifically, the SPRINT working group dedicated to preclinical studies aims to collect and critically analyze preclinical data to provide a clear understanding of the therapeutic potential for specific pathological conditions and the underlying mechanisms in different in vivo models. Systematic and comprehensive reviews of preclinical studies identify research gaps for each animal model and the disease of interest and evaluate therapeutic PnD interventions’ efficacy. SPRINT is focused on PnD applications involving differentiation capabilities into tissue-specific cells or paracrine actions via the release of mediating factors acting on resident and progenitor cells, in line with the body of published benefits in animal models of acute injury and chronic inflammatory diseases.

The COST SPRINT Action (CA17116) published a consensus paper on the human placenta tissue and cell nomenclature [1]. The present protocol aims to: (1) serve as guidelines for the design and reporting of preclinical studies assessing the safety and therapeutic efficacy of PnD; (2) describe the SPRINT consortium’s strategy for a comprehensive search and selection of the significant publications in the field; (3) outline the data extraction used to generate the SPRINT reviews for different therapeutic targets [3,4,5,6]; (4) foster the consensus on the proper study designs, and the knowledge of the therapeutic efficacy of PnD; (5) minimize redundancy in animal experimentation in line with the Replacement, Reduction, and Refinement (3R) principles for more ethical use of animals in research [7]. The focus of the SPRINT reviews of animal models for different diseases and injuries is set on assessing the studies for a clear definition of the research question in terms of population, interventions, comparators, outcomes, and study designs (PICOS) [8]. Accurate description and characterization of the PnD used in animal studies are often suboptimal, and the naming and abbreviations of the PnD types at the authors’ discretion. In the reviews published by the SPRINT consortium, the naming of the PnD types used in the animal studies are harmonized according to the proposed nomenclature for improved comparability.

## 2. Materials and Methods

The SPRINT preclinical studies working group defined a common approach to facilitate and coordinate the collection of published data to be included in the planned disease- or research field-specific reviews. A workflow following the Preferred Reporting Items for Systemic Reviews and Meta-Analyses (PRISMA) guidelines [9,10] was defined as described below.

### 2.1. Search Strategy and Data Mining

A systematic literature search of the PubMed^®^ database [11] was performed and updated on 22 February 2022. After reaching consensus following an open discussion with all SPRINT members, the Boolean search string was designed to maximize the inclusiveness of publications for PnD. With the same scope in mind, additional preclinical search terms for animal models were included (Figure 1A). We excluded non-original research as well as studies using umbilical cord blood or hematopoietic cells from the search string.

### 2.2. Selection of Studies and Inclusion Criteria

The database records identified using the search string were then imported into a Mendeley publication library shared with all COST SPRINT Action members and accessible online or via a locally installed application (Mendeley Desktop version 1.19.8 or Mendeley Reference Manager version 2.66.0, Elsevier B.V., Amsterdam, The Netherlands) (https://www.mendeley.com/) (accessed on February 2022). Eligible studies were selected following a workflow compliant with the PRISMA guidelines, as outlined in Figure 1B. Briefly, more than 12,000 publications were imported into the publication library and assigned to working group members based on the year of publication to verify the inclusion criteria. In the first selection round, the key inclusion criteria were PnD treatments in in vivo models and the evaluation of their efficacy. Only original research studies available as a full text in English were included. In the early years of PnD research, only a few studies were published that met the inclusion criteria, and the definition of PnD was often not accurate or well defined. Therefore, the time frame was restricted to publication years 2004–22 February 2022 (Figure 2). A total of 2259 publications were initially classified as eligible. The aim of the SPRINT preclinical studies working group was to identify diseases or research fields that benefit from PnD therapy. The working group identified diseases or research fields where its members have the necessary expertise to define appropriate outcome measures and judge the eligibility of the studies. The research fields covered by the SPRINT preclinical studies working group include cardiovascular repair and regeneration, neuroprotection and regeneration, in utero prenatal therapy, diabetes or metabolic syndrome, inflammation- and immune-related diseases, oncology, reproduction, bone regeneration, and wound healing. All eligible studies were classified into the defined research fields by the working group leaders. Preclinical working group members with the necessary expertise verified and confirmed or excluded all studies allocated to their respective research fields for the final selection of relevant studies to be included in the meta-analyses or comprehensive reviews. The selection of relevant studies was based on the availability of data from functional tests that mark the current state-of-the-art for assessing therapeutic effects in the respective disease or research field and are significant for the translation of the results into clinical practice.

### 2.3. Data Management

A master relational database (Microsoft Access, Microsoft Corporation, Redmond, WA, USA) including data fields for bibliometric data, publication metrics, and the studies’ experimental data was established (Figure 1C). A Microsoft Excel (Microsoft Corporation) table template representing the same structure and data field formats was provided to all working group members to be filled in with the studies selected in their disease or research fields. Selected publications were either directly imported into the database or from the Microsoft Excel tables. The use of the PubMed identifier (PMID) as a unique identifier allowed for the exclusion of duplicates. The entries were also checked for published corrections to update the original entries and for editorials relating to original studies that were excluded. Publication metrics were used as informational selection criteria to help identify highly cited publications but did not result in the exclusion of publications. InCites Journal Citation Reports (https://jcr.clarivate.com/) (accessed on February 2022), [12] were consulted to retrieve the journal publication metrics, while iCite (https://icite.od.nih.gov/) (accessed on February 2022) (National Institutes of Health, Bethesda, MD, United States of America) [13,14,15,16] and Dimensions (https://app.dimensions.ai/discover/publication) (accessed on December 2020) (Digital Science & Research Solutions Inc., London, UK) [17,18] served for the inclusion of the studies’ individual publication metrics.

### 2.4. Data Extraction

The extraction of the experimental data from the selected studies is finished. For each disease or research field, the experimental data from the selected publications was extracted by working group members or external collaborators with the respective expertise (Figure 1C). If applicable, established naming of the animal models and outcome measures are used. To adhere to the population, interventions, comparators, outcomes, and study designs (PICOS) criteria [8], the recipient (animal) and donor (human or animal) populations are recorded in all details available. Since the analysis of the studies is centered on PnD as the primary intervention, the PnD type and the degree of characterization prior to their administration is registered as published. In a second stage, the PnD type is named according to the proposed consensus nomenclature for PnD [1] to allow for a comparison of their treatment efficacies. The control and experimental groups, including the treatment modalities (vehicle/sham treatment for controls, dosage, single/multiple administrations, time point(s), and route(s) of administration), are described. The primary and secondary outcome measures are noted in conjunction with the respective follow-up period(s) and the time point(s) of analysis. If outcome measures were reported in a graphical format only, the data represented in diagrams are digitized using open resource image processing applications such as WebPlotDigitizer (https://automeris.io/WebPlotDigitizer/) (accessed on February 2022) [19]. The statistical tests used in the studies are also recorded in the master database.

### 2.5. Data Synthesis

Systematic reviews are carried out to assess the treatment efficacy and the clinical benefit of PnD by comparing the outcomes measured in clinical scores in non-treated vs. treated animals, when possible (Figure 1D). All types of PnD used in a specific disease model are compared to indicate the suitability of the different PnD types as a treatment for a given disease or injury. Likewise, the routes of administration, the dosages, and the frequency of administration are compared by measuring the differences in clinical scores as a readout for improvement. The quality of preclinical studies is typically rather heterogeneous. Therefore, a risk of bias assessment is evaluated using tools such as the SYRCLE’s risk of bias tool for animal studies [20], the revised Cochrane’s risk of bias evaluation tool (RoB-2) [21] or ROBINS-I tool for non-randomized studies [22] to indicate low, high or unclear risk of bias for the description of the host animals’ characteristics, randomization process, blinding of group allocation and outcome measurement and completeness of outcome data reporting [23,24].

## 3. Discussion

It is the aim of the International Network for Translating Research on Perinatal Derivatives into Therapeutic Approaches (SPRINT, CA17116) to provide independent and scientifically sound knowledge and guidelines to foster the safe and efficient therapeutic use of PnD. To our knowledge, a systematic review of preclinical studies comparing all types of PnD for a variety of diseases and research fields was not conducted prior to SPRINT initiatives. 

From this extensive work of literature search and selection and data extraction, several systematic and comprehensive reviews are published already. These publications gathered and analyzed the existing literature on the application of preclinical studies on different fields, namely cutaneous wound healing [5], ovarian diseases [3], oncology [6], and ophthalmology [4]. All of these publications used the common search and data extraction strategy described here, with additional, disease-specific refinements as described in detail in the respective publications. The common reporting scheme included the description of the PnD type, the in vivo dosage, the time points and route of application, the animal species, the disease model and the outcomes. In general, the designs of preclinical studies are very heterogeneous with low numbers of studies with comparable design and analysis, which makes it difficult to compare study results. PnD therapy proved to be beneficial in all investigated disease models. The proposed mode of action is involving paracrine pathways, such as the reduction in oxidative stress and apoptosis^3^ or anti-tumor and immunomodulatory responses [6]. In most disease models, however, a clear statement on the most effective PnD type, dosage, time points or mode of application could not be made due to the lack of direct comparisons within the same studies. Another limitation to the future translation of preclinical results into clinical therapies is the lack of large animal models or the assessment of the in vivo biodistribution and the fate of applied PnD. As a consequence, the SPRINT consortium made recommendations on the reporting of PnD isolation and characterization, on the disease models with standardized time points for analysis and type of readouts and on the requirement to conduct in vitro functional tests to improve the reproducibility and power of preclinical studies [4,5].

To validate future therapies, animal models are still widely regarded as the gold standard for the evaluation of therapies in diseases with a multifactorial pathology or involving complex organs. The meta-analysis of preclinical studies contributes to the 3R principle and will, therefore, have long-term social and ethical impacts. To further improve the long-term use of future preclinical studies, we recommend filling out the Compliance Questionnaire provided by the Animal Research: Reporting of In Vivo Experiments (ARRIVE, https://arriveguidelines.org/) (accessed on February 2023) guidelines [25].

The host animal breed or strain (including information on mutant alleles and immune status) is generally well documented, but information on the human donor ethnicity, age, gestational age, etc., is often missing or not well described. While the collection of PnD tissue under an anonymized donation scheme without disclosure of personal data is compliant with the Code of Ethics, the documentation of general patient-related data as stated above would be beneficial for future meta-analyses taking these parameters into account.

Obviously, there is a significant amount of variability between diseases for the choice of primary and secondary experimental outcome measures. Within research fields, there is often a broad spectrum of analytical and behavioral tests used to assess the efficacy of the treatments. Well-defined and standardized animal models are required to compare outcome measures between studies. Human diseases and injuries come in various distinct subtypes, while animal models often represent only the pathologies’ primary aspect(s). Preclinical models designed for the best possible representation of the human disease might deviate from the standard models that would allow for an adequate meta-analysis of different treatments. This conflict of interest is challenging to overcome. General statements made for the treatment efficacy of types of PnD, routes, time points and frequencies of administration, and dosages might hold true for standardized models but have to be carefully re-evaluated to find the best treatment for a specific pathology or patient population. Nevertheless, the International Network for Translating Research on Perinatal Derivatives into Therapeutic Approaches (SPRINT, CA17116) firmly believes that in the future, therapeutic applications of PnD and the knowledge gain of their therapeutic potential and underlying mechanisms will clearly profit from an in-depth, comprehensive, and evidence-based analysis of all data available from basic science, preclinical, and clinical studies.

## Figures and Tables

**Figure 1 mps-06-00045-f001:**
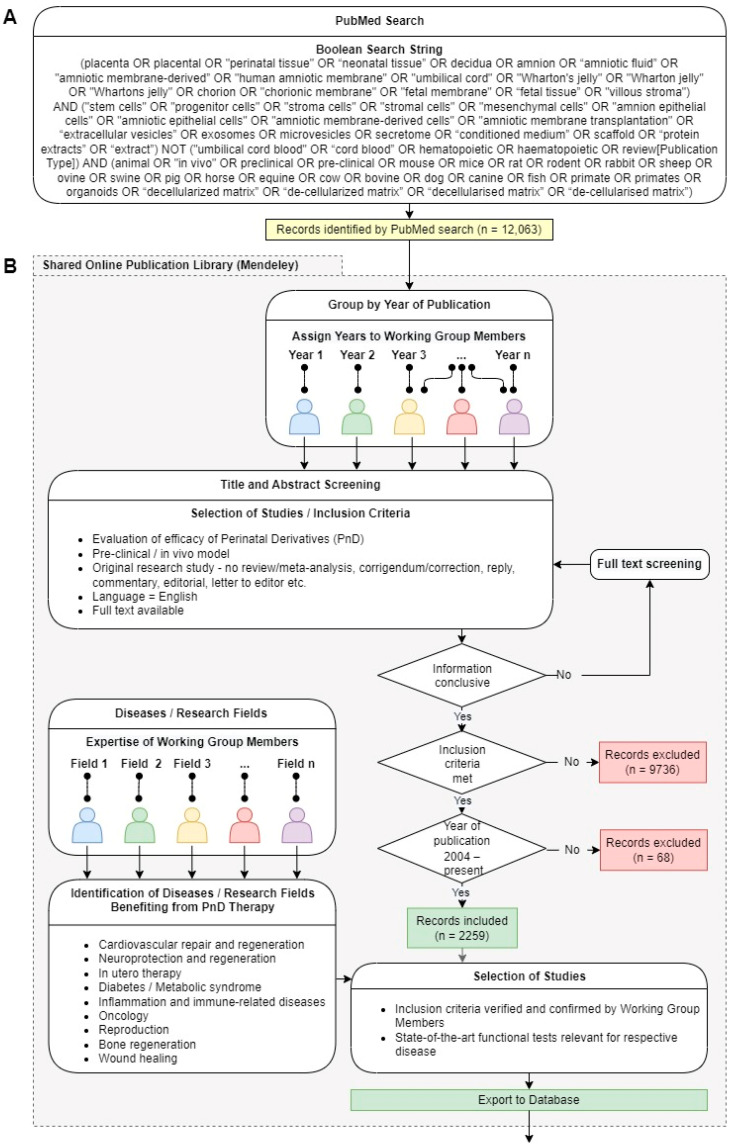
Boolean search string used to identify preclinical studies using perinatal derivatives (PnD) as a therapeutic approach (**A**). Workflow for the selection of studies using the internally shared Mendeley publication library (**B**). Workflow for the data collection of selected studies indicating data containers for publication metrics, nomenclature guideline, and data digitalization tool (**C**). Common denominators for the meta-analyses in the different research fields (**D**).

**Figure 2 mps-06-00045-f002:**
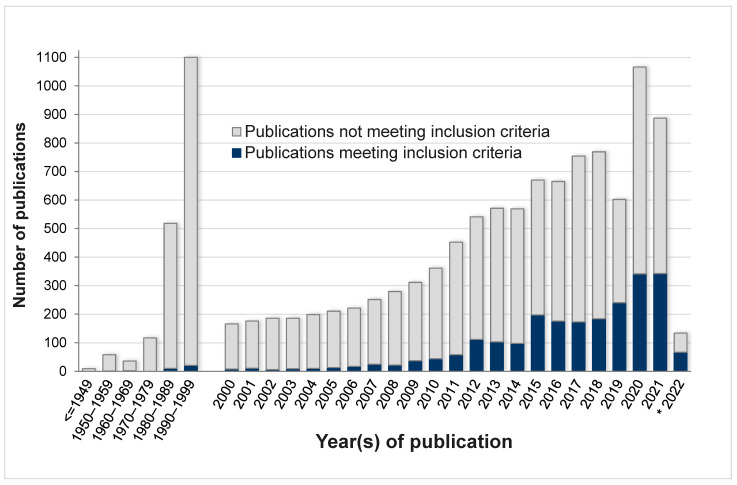
Number of publications identified in PubMed through the Boolean search string (see Figure 1A) and classified into publications meeting inclusion criteria (dark blue boxes) vs. publications not meeting inclusion criteria (light grey boxes) as defined in Figure 1B, by decade or year of publication. Records starting from 2004 were considered for inclusion in the database. * 2022 includes publications until 22 February 2022.

## Data Availability

Access to the publication libraries will be granted on justified request.

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
