# Peer review of "Guidelines to Analyze Preclinical Studies Using Perinatal Derivatives"

_mps, 2023, doi:10.3390/mps6030045_

Round 1

Reviewer 1 Report

1. This is an important topic to review: preclinical studies using perinatal derivatives. The suggested study is very detailed.

2. While it is understandable this is a Protocol, the aim/aims of the study is muddled - not clear. or If there are multiple aims, they need to be logical and listed systematically.

Is it:  a. to derive Guidelines to analyse preclinical studies using perinatal derivatives or 

b. to describe the publication search and data mining, extraction, and synthesis strategies employed to collect and prepare the published data selected for meta-analyses and reviews of the efficacy of PnD therapies for different diseases and injuries or 

c. to provide standards to assess the safety and clinical benefit of PnD and to minimize redundancy in the use of animal models following the 3R principles for animal experimentation. 

d. to provide independent and scientifically sound knowledge and guidelines to foster the safe and efficient therapeutic use of PnD.

3. Fig 2- the definitions of relevant or irrelevant publications are not clear

4. It is not clear what has been done/published and what have not been done or will be done with this protocol from the paragraph - 'To our knowledge, a systematic review of preclinical studies comparing all types of PnD for a variety of diseases and research fields was not conducted prior to SPRINT initiatives.  From this extensive work of literature search and selection and data extraction, several systematic and comprehensive reviews are published already. These publications gathered and analyzed the existing literature on the application of preclinical studies on different fields, namely cutaneous wound healing, ovarian diseases, oncology and ophthalmology. Other reviews on further relevant research fields are being prepared to be promptly published in relevant peer reviewed scientific journals.'

5. Also were there any other protocols that produced the published reviews? or just one encompassing protocol?

Author Response

Dear Reviewer 1,

We thank the revision made to our paper “Guidelines to analyse preclinical studies using perinatal derivatives” and the opportunity for being revised. We have carefully contemplated the concerns raised by the Reviewer 1 and changed the manuscript in accordance to the suggestions raised.

We thank the learned and critical comments on our paper. Below you can find the specific answers to the remarks, written in blue.

Please see the attachment to see all the alterations made.

Responses to Reviewer 1 (in blue)

  1. This is an important topic to review: preclinical studies using perinatal derivatives. The suggested study is very detailed.

Answer: We thank the reviewer the comment on the pertinence of our manuscript.

  1. While it is understandable this is a Protocol, the aim/aims of the study is muddled - not clear. or If there are multiple aims, they need to be logical and listed systematically.

Is it:

  1. to derive Guidelines to analyse preclinical studies using perinatal derivatives or
  2. to describe the publication search and data mining, extraction, and synthesis strategies employed to collect and prepare the published data selected for meta-analyses and reviews of the efficacy of PnD therapies for different diseases and injuries or
  3. to provide standards to assess the safety and clinical benefit of PnD and to minimize redundancy in the use of animal models following the 3R principles for animal experimentation.
  4. to provide independent and scientifically sound knowledge and guidelines to foster the safe and efficient therapeutic use of PnD.

Answer: Thank you for this valuable input. We have now better specified the aims of the present manuscript in the introduction (lines 96-115).

  1. Fig 2- the definitions of relevant or irrelevant publications are not clear

Answer: For clarity, the stacked bars have now been labelled “publications not meeting inclusion criteria” and “publications meeting inclusion criteria”, respectively. The figure caption was adapted accordingly with a reference to Figure 1B.

  1. It is not clear what has been done/published and what have not been done or will be done with this protocol from the paragraph - 'To our knowledge, a systematic review of preclinical studies comparing all types of PnD for a variety of diseases and research fields was not conducted prior to SPRINT initiatives. From this extensive work of literature search and selection and data extraction, several systematic and comprehensive reviews are published already. These publications gathered and analyzed the existing literature on the application of preclinical studies on different fields, namely cutaneous wound healing, ovarian diseases, oncology and ophthalmology. Other reviews on further relevant research fields are being prepared to be promptly published in relevant peer reviewed scientific journals.'

Answer: We agree that this might be confusing. Therefore, we now only listed the already published manuscripts that used the present guidelines for the selection of publications and the data extraction and analysis. While at least one other publication is currently being prepared for publication, it is still unclear whether it will be accepted. Therefore, the last sentence of the above mentioned section was deleted.

  1. Also were there any other protocols that produced the published reviews? or just one encompassing protocol?

Answer: The search protocol was the same for the different reviews already published and cited in this manuscript. To clarify, we added the following sentence: “All of these publications used the common search and data extraction strategy described here, with additional, disease-specific refinements as described in detail in the respective publications.” (lines 249-252).

Reviewer 2 Report

05.April.2023

Manuscript ID: mps-2259530, entitled “Guidelines to analyse preclinical studies using perinatal derivatives”.

Comments to the Authors

Thank you for the opportunity to review this manuscript which aims to provide systematic and comprehensive reviews of preclinical studies for the understanding of the therapeutic potential and mechanisms of perinatal derivatives in diseases and injuries that benefit from perinatal derivatives therapy.

In the last two decades, there have been significant advances in the research and understanding of the biology of the placenta and its derivatives. Many  studies and initial clinical trials reported that perinatal derivatives may represent important tools for restoring tissue damage or promoting regeneration and repair of the tissue microenvironment. But this topic still in research and need more evidence based.

I suggest going into more detail in the introduction and the discussion regarding the biological effect presented in the introduction, because it is not really clear what the findings of these studies were.

Author Response

Dear Reviewer,

We thank the revision made to our paper “Guidelines to analyse preclinical studies using perinatal derivatives” and the opportunity for being revised. We have carefully contemplated the concerns raised by the Reviewer 2 and changed the manuscript in accordance to the suggestions made.

We thank the learned and critical comments on our paper. Below you can find the specific answers to the remarks, written in blue.

Please see the attachment to see all the alterations made.

Responses to Reviewer 2 (in blue)

I suggest going into more detail in the introduction and the discussion regarding the biological effect presented in the introduction, because it is not really clear what the findings of these studies were.

Answer: We thank the Reviewer for the pertinent suggestion. To clarify, we have now better specified the aims of the present manuscript in the introduction (lines 96-115), and summarized the findings of the reviews published by the SPRINT consortium in the discussion (lines 252-267).

Round 2

Reviewer 1 Report

The authors have addressed my comments.